# Scalable total synthesis of (+)-aniduquinolone A and its acid-catalyzed rearrangement to aflaquinolones

Feng-Wei Guo[1,2,6], Xiao-Feng Mou[1,3,6], Yong Qu[1,2], Mei-Yan Wei[1], Guang-Ying Chen[4], Chang-Yun Wang[1,2], Yu-Cheng Gu [5] & Chang-Lun Shao [1,2✉]

The strong antibacterial, antiviral and anticancer activities demonstrated by quinolones make them promising lead structures and important synthetic targets for drug discovery. Here, we report, to the best of our knowledge, the first scalable total synthesis of antiviral (+)-aniduquinolone A, possessing a 3,4-dioxygenated 5-hydroxy-4-aryl-quinolin-2(1H)-one skeleton. This synthetic strategy explores E-stereoselective Horner–Wadsworth–Emmons (HWE) olefination as the key step to assemble isopropenyl substituted tetrahydrofuran onto the 3,4-dioxygenated 5-hydroxy-4-aryl-quinolin-2(1H)-one core, which is built by highly diastereoselective intramolecular aldol reaction. Moreover, two sets of stereoisomers of aniduquinolone A with substantially overlapping NMR data were synthesized completely and assigned unambiguously by comprehensive analysis of both their spectroscopic and X-ray diffraction data. Unexpectedly, aflaquinolones A, C, and D that feature different 2,4-dimethyl cyclohexanone moieties were transformed successfully from (+)-aniduquinolone A by treating with TFA. The methodology delineated herein can be applied broadly to the synthesis of natural alkaloids containing the core structure of 3,4-dioxygenated 5-hydroxy-4-aryl-quinolin-2(1H)-one.

[1] Key Laboratory of Marine Drugs, The Ministry of Education of China, School of Medicine and Pharmacy, Ocean University of China, Qingdao 266003, China. [2] Laboratory for Marine Drugs and Bioproducts, Pilot National Laboratory for Marine Science and Technology (Qingdao), Qingdao 266200, China. [3] School of Pharmacy, Yantai University, Yantai 264005, China. [4] College of Chemistry and Chemical Engineering, Hainan Normal University, Haikou 571158, China. [5] Syngenta Jealott's Hill International Research Centre, Bracknell, Berkshire RG42 6EY, UK. [6] These authors contributed equally: Feng-Wei Guo, Xiao-Feng Mou. ✉email: shaochanglun@163.com

Natural products (NPs) have been holding the best options for finding bioactive scaffolds and serve as an inspiration for the development of new pharmaceuticals[1–3]. Approximately 50% of small-molecule drugs are derivates of or inspired by NPs[1]. Quinolones play a pivotal role in drug development as one of the key pharmacophore in modern drug design[4–6]. The 3,4-dioxygenated 5-hydroxy-4-aryl-quinolin-2(1*H*)-one alkaloids[7] constitute a relatively new and steadily growing family of natural products with diverse biological properties, including cytotoxic[8–10], antiviral[11,12], antioxidant[13], antifouling[14], and anti-inflammatory[15] activities. The structures of few representatives including aniduquinolones A (**1**) and B (**2**)[16], yaequinolone F (**3**)[17,18], aflaquinolones A–D (**4–7**)[14,19], (+)-22-*O*-(*N*-Me-L-valyl)-aflaquinolone B (**8**) and (+)-22-*O*-(*N*-Me-L-valyl)-21-*epi*-aflaquinolone B (**9**)[11] are depicted in Fig. 1. These alkaloids possess a 3,4-dioxygenated 5-hydroxy-4-aryl-quinolin-2(1*H*)-one core fused to C10 isoprenoid motif, which is rare in nature[7].

Despite possessing promising biological properties and synthetically attractive motifs, relatively few of these alkaloids especially those with a C-6 side chain containing an isoprenyl derived (C5 or C10) unit have been prepared by total synthesis (Fig. 2a)[20–25]. Simonetti and co-workers developed a convenient way to install 6-propenyl side chain by Claisen rearrangement of 5-*O*-allyl in the heterocycle[21]. Similar reaction was also used in the construction of unsaturated pyran fragments in yaequinolones J1 and J2[26] by Vece and co-workers[22], as well as the synthesis of aniduquinolone C and peniprequinolone by Christmann's group[23]. Recently, Christmann's group[24] developed the tandem Knoevenagel electrocyclization to further optimize the assembly of pyran fragments in yaequinolones J1 and J2. In 2021, the group of Fernández-Ibáñez[25] successfully synthesized yaequinolone-related natural products by late-stage C−H olefination to introduce C-6 side chains. Although the feasibility of generating the above members of this family on a milligram scale was demonstrated over the past decade, the ultimate challenge of procuring large quantities of these alkaloids has yet to be met. To the best of our knowledge, there have been no report to date for the total synthesis of aniduquinolone A (**1**).

Herein, we present an efficient and scalable approach to construct this family of natural products by introducing an *E*-stereoselective Horner–Wadsworth–Emmons (HWE) olefination to link an isoprenoid motif with 3,4-dioxygenated 5-hydroxy-4-aryl-quinolin-2(1*H*)-one core. Furthermore, we describe the first total synthesis of (+)-aniduquinoline A (**1**) on a gram scale and its unexpected acid-catalyzed rearrangement to form the natural aflaquinolones (**4–6**) possessing different 2,4-dimethyl cyclohexanone moieties.

## Results and discussion

**Retrosynthetic analysis.** The retrosynthetic analysis of (+)-aniduquinoline A (**1**) in a collective fashion is shown in Fig. 2b. The focus of our synthetic strategy is scalability with the aim to prepare reasonable amount of **1**. The 3,4-dioxygenated 5-hydroxy-4-aryl-quinolin-2(1*H*)-one core of target molecules was envisioned to be obtained rapidly by a diastereoselective intramolecular glycolate aldolization[20] of *N*-glycolated 2-amino-benzophenone **10**. We speculated that this precursor could be derived from the reduction and amidation of benzophenone **11**, resulting from the *E*-stereoselective HWE reaction[27] of phosphonate **12** and aldehyde **13**. It should be mentioned that such strategy towards the construction of *E* olefins in this family of natural products would improve the efficiency of their synthesis dramatically. The phosphonate **12** could be furnished from the radical-mediated bromination and Arbuzov reaction of benzophenone **14**, which could readily be prepared from the commercially available 2-methyl-5-nitrophenol **15**. Furthermore, the substituted *trans*-tetrahydrofuran aldehyde **13** could arise from diol **16**, which in turn could be accessed from neryl acetate **17**[28].

**Scalable total synthesis of (+)-aniduquinolone A.** The synthesis started with the commercially available 2-methyl-5-nitrophenol **15**, which was submitted to a Duff formylation[21] with hexamethylenetetramine (HMTA) in CF$_3$COOH at 90 °C for 12 h, affording the expected aldehyde **18** in 70% yield (Fig. 3). Subsequently, compound **19** was prepared via the reaction of phenylmagnesium bromide with aldehyde **18** (62.0 g scale). No branched side product was detected during this transformation. Pyridinium chlorochromate (PCC) oxidation[29] of secondary alcohol **19** afforded the corresponding ketone **14**. The structure of **14** was further confirmed by single-crystal X-ray analysis (Supplementary Fig. 1 and Supplementary Data 2). The unprotected hydroxyl group in **14** was converted into a MOM ether derivative. The radical-mediated bromination of

**Fig. 1 Examples of bioactive quinolones.** Representatives of the 3,4-dioxygenated 5-hydroxy-4-aryl-quinolin-2(1*H*)-one alkaloids.

Figure labels:
(+)-aniduquinolone A (**1**)
(19*R**, 22*R**): (+)-aniduquinolone B (**2**)
(3*S**, 4*S**): (+)-yaequinolones F (**3**)
(19*R*, 21*S*): (+)-aflaquinolone A (**4**)
(19*S*, 21*R*): (-)-aflaquinolone C (**5**)
(19*S**, 21*S**): (-)-aflaquinolone D (**6**)
(+)-aflaquinolone B (**7**)
(21*S*): (+)-22-*O*-(*N*-Me-L-valyl)-aflaquinolone B (**8**)
(21*R*): (+)-22-*O*-(*N*-Me-L-valyl)-21-*epi*-aflaquinolone B (**9**)

**Fig. 2 Synthetic strategy. a** Previous synthetic routes of the 3,4-dioxygenated 5-hydroxy-4-aryl-quinolin-2(1H)-one containing alkaloids. **b** Retrosynthesis of (+)-aniduquinolone A (**1**).

methoxymethyl (MOM)-protected **14** in the presence of NBS with a catalytic amount of AIBN failed to produce any desired product. Gratifyingly, the radical-mediated bromination of **20** under basic condition furnished bromoester that was reacted directly with an excess of triethyl phosphite and delivered the desired phosphonate **21** in 72% yield (15.0 g scale)[30].

With ample amounts of key intermediate **21** in hand, we then focused our attentions on the asymmetric synthesis of the important aldehyde **13**. To achieve this goal, the known diol **16** was obtained by stereospecific elimination-cyclization of 1-iodomethyl-1,5-bis-epoxides, and its structure and absolute configuration were assigned based on published literature[28]. As we expected, the oxidation of intermediate **16** by $NaIO_4$ afforded the desired aldehyde **13** (5.0 g scale).

With key fragments **13** and **21** in place, their assembly into (+)-aniduquinolone A (**1**) was embarked. The stereoselective HWE reaction of aldehyde **13** with phosphonate **21** proved to be more problematic than anticipated. While initial trials using phosphonate having the acetoxy group **21** with aldehyde **13**, failed to produce any desired product and only the product with

protecting group removed was obtained. This result suggested that the protection strategy proved to be challenging because a robust protecting group was necessary during the synthetic route. At this juncture, conditions for the HWE reaction[27] to construct the E olefins needed to be developed. Reaction conditions for the key olefination were investigated using **21**, **21a** and **12** as substrates (Table 1). Fortunately, we discovered eventually that MOM protected phosphonate **12** underwent olefination smoothly and afforded the desired product **11** in 78% yield with no detection of the corresponding Z-isomer (Table 1, entry 3, 3.9 g scale, E/Z = 100:0).

With the scalable formation of **11** secured, subsequent reduction of the nitro group was performed in the presence of Fe powder[22] (3.9 g scale). The resulting aromatic amine **22** together with a diastereomer (1:1.2) was obtained. We anticipated that the desired product **22** could be produced in higher yield at low temperature while reducing the diastereomer[31]. Working on this hypothesis, the initial trials were screened by varying the reaction temperature from 80 to 30 °C. To our delight, when the reaction temperature was at 30 °C, the desired aromatic amine **22**

**Fig. 3 Total synthesis of (+)-aniduquinolone A (1).** Reagents and conditions: **a** HMTA (1.2 equiv), TFA, 90 °C, 12 h, 70%; **b** phenylmagnesium bromide (2.0 equiv), THF, −20 °C, 2 h, 91%; **c** PCC (1.0 equiv), CH₂Cl₂, 0 °C, 4 h, 80%; **d** H₂SO₄, (CH₃CO)₂O/CH₂Cl₂ (1:2), 0 °C to rt, 12 h, 89%; **e** NBS (1 equiv), AIBN (0.05 equiv), CCl₄, 80 °C, 24 h; **f** P(OCH₂CH₃)₃ (4 equiv), 1,4-dioxane, 120 °C, 8 h, 72% (two steps); **g** K₂CO₃ (1.2 equiv), MeOH, rt, 1 h; **h** MOMCl (1.5 equiv), DIPEA (2 equiv), 0 °C to rt, 2 h, 76%(two steps); **i** NaH (1.2 equiv), THF, 0 °C, 2 h, 78%; **j** Fe (4.0 equiv), NH₄Cl (2.0 equiv), EtOH/H₂O (1:1), 30 °C, 6 h, 92%; **k** DIPEA (2.0 equiv), methoxyacetyl chloride (1.5 equiv), CH₂Cl₂, 0 °C, 2 h; **l** KOtBu (10.0 equiv), THF, 0 °C, 1 h, 80% (d.r. 1:1, two steps); **m** 3 M HCl, THF, rt, 30 min, 54%; **n** NaIO₄ (2.0 equiv), THF/H₂O (2:1), rt, 30 min, 73%. HMTA hexamethylenetetramine, TFA trifluoroacetic acid, THF tetrahydrofuran, PCC pyridinium chlorochromate, NBS N-bromosuccinimide, AIBN azodiisobutyronitrile, MOMCl methoxymethyl chloride, DIPEA diisopropylethylamine, KOtBu potassium tert-butylate.

was isolated in 92% yield (3.3 g scale) with very high diastereoselectivity (d.r. > 20:1). Treatment of the resulting aromatic amine **22** with methoxyacetyl chloride afforded the amide **10**. At this stage, compound **10** bears all the necessary carbon atoms present in (+)-aniduquinolone A (**1**).

We then moved to the stage to complete the total synthesis. We investigated an intramolecular aldol reaction to forge the 3,4-dioxygenated 5-hydroxy-4-aryl-quinolin-2(1H)-one core. The crude amide **10** (not purified) was directly treated with potassium *tert*-butoxide in THF at 0 °C to initiate the aldol cyclization and provided diastereomeric **23** and **23a** in a ratio of 1:1 and in 80%

yield over two steps (3.0 g scale)[20–25]. The more stable Z-enolate is presumably responsible for the observed diastereoselectivity[32]. After many attempts, the two diastereoisomeric aldol products **23** and **23a** were separated successfully from one another by recrystallization in MeOH. Finally, the deprotection of MOM group of **23** with 3 M HCl afforded (+)-aniduquinolone A (**1**) (54% yield of isolated product). The synthetic (+)-aniduquinolone A gave spectral characteristics (¹H- and ¹³C-NMR spectroscopy and HRMS data) consistent with those of the natural occurring (+)-aniduquinolone A, and its optical rotation is also in agreement perfectly with that of the natural product (synthetic:

**Table 1 Optimization of reaction conditions.**

| Entry | Substrate | Conditions | Yield of 11 (%) |
|---|---|---|---|
| 1 | **21** | NaH, THF, °C, 2 h | ND[a] |
| 2 | **21a** | NaH, THF, 0 °C, 2 h | ND |
| 3 | **12** | NaH, THF, 0 °C, 2 h | 78[b, c] |
| 4 | **12** | KO$^t$-Bu, THF, −78 °C to 0 °C, 2 h | 52 |
| 5 | **12** | $n$-BuLi, THF, −78 °C to RT, 6 h | Trace |
| 6 | **12** | LHMDS, THF, −78 °C to 0 °C, 2 h | 75 |
| 7 | **12** | NaOMe, DMF, 0 °C, 12 h | 19 |
| 8 | **12** | NaH, DMF, 0 °C to RT, 12 h | Trace |
| 9 | **12** | NaH, toluene, 0 °C to RT, 2 h | 45 |

[a]Not detected.
[b]Gram scale.
[c]$E/Z = 100$:0.

$[\alpha]_D^{20} = +56.0$ ($c = 0.7$ in MeOH); natural: $[\alpha]_D^{20} = +50.0$ ($c = 0.1$ in MeOH)[16]. Finally, the structure of synthetic material was confirmed by X-ray single crystal analysis (Supplementary Fig. 2 and Supplementary Data 1).

**Two sets of interesting diastereoisomers (compounds 1/26, 25/27) with substantially overlapping NMR data.** We envisioned that the aldol reaction would create a pair of intermediates (**23** and **23a**) with identical absolute configurations in tetrahydrofuran fragment and mirrored absolute configurations in quinolinone core. Ultimately, the desired **1** and *ent*-**26** would be obtained by deprotection of **23** and **23a**, respectively. However, treating **23** with 3 M HCl delivered a 1.4:1 ratio of (+)-aniduquinolone A (**1**) and its C-19 or C-22 epimer (**25 or 27**) in 95% yield (Fig. 4). The same transformation was also observed in the deprotection of **23a**. It should be mentioned that the relative configurations of **25** and **29** could be assigned directly by the selective NOE experiments, but their absolute configurations would still be unresolved. Recently, aniduquinolone A (**1**) and its inseparable C-22 epimer aniduquinolone D as mixtures were reported from the endophytic fungus *Aspergillus versicolor* strain Eich.5.2.2[33]. It should be emphasized that the absolute configuration of aniduquinolone D was arbitrarily determined by examination of ROESY spectrum and comparison with the reported NMR spectral data of aniduquinolone A in a mixture.

At this juncture, we attempted to synthesize aniduquinolone D with 3*S*, 4*S*, 19*R*, 22*S* configurations and to determine the absolute configurations of **25** and **29**. The same end game was executed with similar yields by using the aldehyde **13a**, and compounds **24** and **24a** were successfully constructed. Deprotection of the MOM group furnished compounds **27**, **31**, and their epimers **26** and **30**. To our surprise, the two compounds **25** and **27** have different retention times and substantially overlapping NMR data (Fig. 5). As far as we know, only a few edificatory examples of diastereoisomers with substantially overlapping [1]H and [13]C NMR data were reported[34–38]. The absolute configurations of C-3 and C-4 in quinolinone fragment could be determined by CD spectrum (Supplementary Figs. 4–6) regardless of the configuration of the furan fragment[19]. The relative configurations of the

substituted tetrahydrofuran unit had been assigned directly by the selective NOE experiments. Thus, the absolute configurations of all synthesized compounds were confirmed. Fortunately, the structures of **1** and **29** were confirmed unambiguously by X-ray crystallographic analysis (Supplementary Figs. 2 and 3 and Supplementary Data 1 and 3). Finally, we proposed that **25** should be the true structure of aniduquinolone D.

In fact, we had noticed that MOM-protected intermediates **23** and **23a** possessed substantially overlapping [1]H/[13]C NMR data and the same *J*-coupling patterns, indicated a weak effect on the chemical shift caused by configurational changes of C-3 and C-4. The acid-inspired transformation led to two sets of diastereomeric compounds (**1/26** and **25/27**) with the same nuclear magnetic properties, which made the absolute configuration assignment extremely challenging. Having so many closely related stereoisomers in hand afforded unique opportunities for comparisons. How similar are these isomers? How can they be differentiated? We address these questions herein by comparing spectral and physical data of these stereoisomers with each other and with reported natural products data in literatures[14,16].

Resonances in the [1]H NMR and [13]C NMR spectra of obtained stereo structures matched closely with each other (Fig. 5 and Supplementary Figs. 7 and 8). The compounds could be divided into two groups (compounds **1/26** and **25/27**) based on the relative configurations of substituted tetrahydrofuran unit, which caused subtle differences in their [1]H NMR data. Clear distinctions of the chemical shifts of two *trans* double bond protons (H-17 and H-18), an oxymethine proton (H-22) and protons of two methylenes (H$_2$-20 and H$_2$-21) were observed in both groups. Of note, compared with **1** and **26**, the chemical shifts of H-17, H-18, and H-22 (the proton which has NOE correlation with CH$_3$-26) in **25** and **27** moved downfield. By analysis of the chemical shift differences, the configurations of the substituted tetrahydrofuran units could be determined rapidly and conveniently.

With so many closely related stereoisomers (**1**, **25–31**) and analogues (**23**, **23a**, **24**, and **24a**) in hand, an efficient [1]H NMR spectroscopic approach for determining the relative configurations of the substituted tetrahydrofuran was established. Compounds **1**, **25**, **26**, and **27** were taken as an examples to give a subsequent conclusion. The values of $\Delta\delta_{\text{H-20}}$ of **1**, **25**, **26**, and **27**

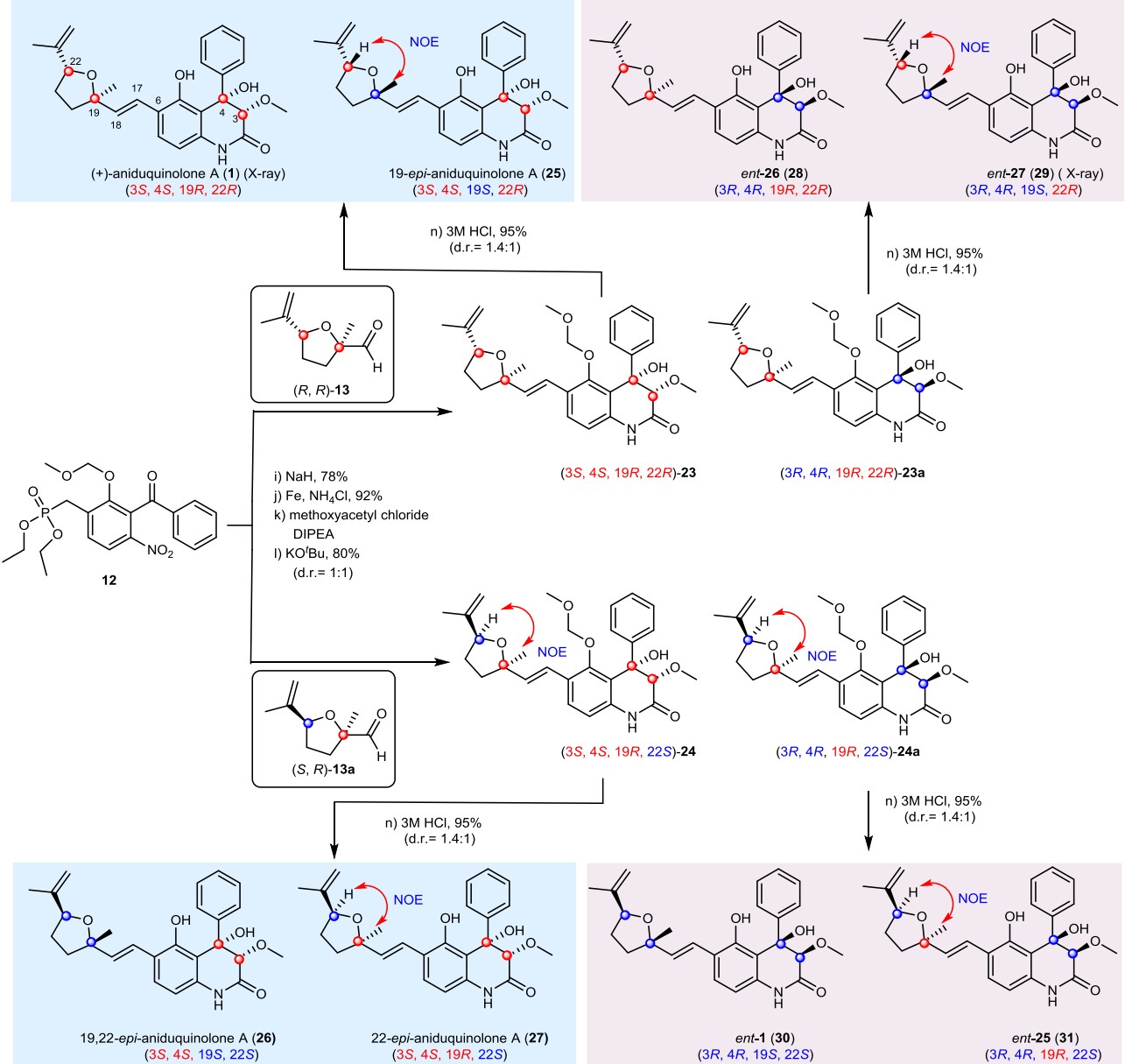

**Fig. 4 The syntheses of (+)-aniduquinolone A (1) and its stereoisomers.** With **12** as the common intermediate, the syntheses of **1** and **25–31** were achieved.

were shown (Supplementary Fig. 9). The H-22/CH₃-26 *trans* compounds (**1** and **26**) showed a large value ($\Delta\delta \geq 0.18$ ppm); the H-22/CH₃-26 *cis* compounds (**25** and **27**) showed a small value ($\Delta\delta \leq 0.13$ ppm). Moreover, compared with compounds **1** and **26**, the chemical shifts of H-17, H-18, and H-22 in compounds **25** and **27** moved downfield. All these NMR characteristics delineated here were instructive to configurational determination of (+)-aniduquinolone A (**1**) and its stereoisomers.

The case of compounds **1**/**26** and **25**/**27** is thought-provoking. The two entirely segregated stereoclusters in aniduquinolones connected by the five-carbon bridge are far away from each other so that the steric and stereoelectronic interactions between them are entirely negligible[34]. The phenomenon of the molecules with substantially overlapping NMR data that may be also diastereoisomers can be explained well. In the structural elucidation, especially for the configurational assignments of molecules with multi-stereoclusters, it is good to be cautious when encountering the substantially overlapping NMR data.

**Synthesis of aflaquinolones A, C, and D by the unexpected acid-catalyzed rearrangement.** The transformation from **1** to 19-*epi*-aniduquinolone A (**25**) with 3 M HCl indicated that (+)-aniduquinolone A is unstable under acidic condition. Intrigued by this observation, we carried out additional studies to understand this transformation. To our surprise, when (+)-aniduquinolone A (**1**) was treated with TFA in CH₂Cl₂ at room temperature, aflaquinolone A (**4**), aflaquinolone C (**5**), aflaquinolone D (**6**), and 19, 21-*epi*-aflaquinolone D (**32**) were obtained in 70% combined yield (Fig. 6). Surprisingly, despite that aflaquinolone A (**5**) and aflaquinolone C (**6**) are epimers, all trials to separate them into individual component using different solvent systems by normal and reversed chromatography deemed impossible. Finally, the two diastereoisomers were separated by using chiral chromatography. The spectroscopic data of our synthetic samples were in good accordance with reported data of natural aflaquinolones A, C, and D (**4**–**6**)[19].

To reach an understanding of unexpected acid-catalyzed rearrangement, a mechanism was postulated in Fig. 6. It was

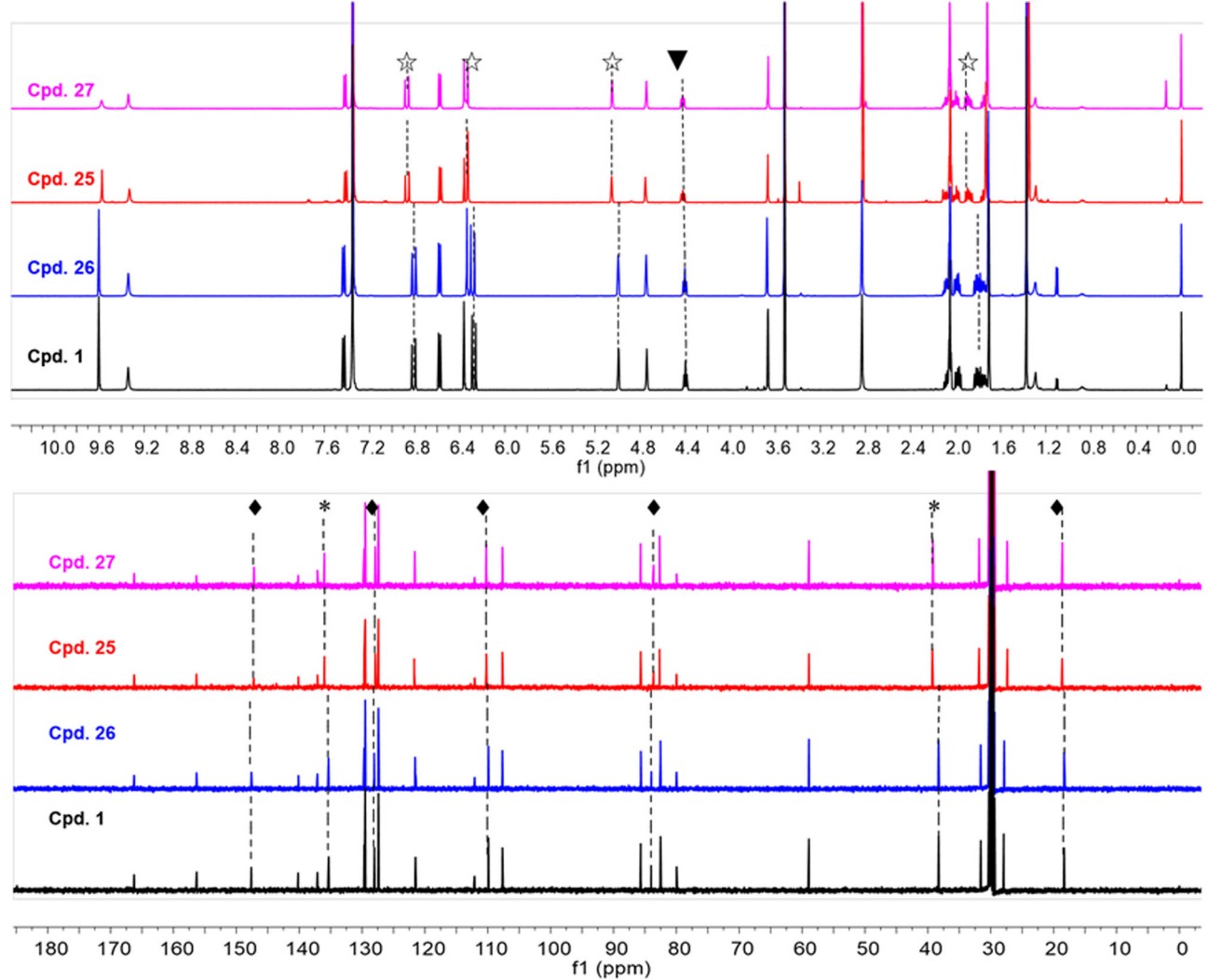

**Fig. 5 Compounds 1 and 26 (or 25 and 27) have substantially overlapping NMR data.** The ¹H NMR spectra of **1**, **25**, **26** and **27** from δ H 0 to 10 ppm; the ¹³C NMR spectra of **1**, **25**, **26** and **27** from δ C 0 to 180 ppm. The proton signals with obvious deviations between **1/26** and **25/27** are labelled: ☆, 0.1 ppm > |Δ δ| > 0.05 ppm; ▼, 0.05 ppm > |Δ δ| > 0.01 ppm; the carbon signals with obvious deviations between **1/26** and **25/27** are labelled: *, 1.0 ppm > |Δδ| > 0.5 ppm; ◆, 0.5 ppm > |Δδ| > 0.1 ppm.

hypothesized that, under acidic conditions, (+)-aniduquinolone A (**1**) underwent a ring-opening reaction to form cationic intermediate B. On the one hand, cyclization of B took place to give 19-*epi*-aniduquinolone A (**25**) with an antiorientation between the methyl group at C-19 and the isopropeny group at C-22 by path A. The transformation delivered a 1.4:1 ratio of (+)-aniduquinolone A (**1**) and its C-19 epimer (**25**). On the other hand, the epoxidation of B then gave epoxide intermediate D by path B. Subsequent ring-opening of cyclohexene oxide afforded enol E, which itself underwent enol-keto tautomerism to produce aflaquinolones. To the best of our knowledge, the unprecedented transformation on such molecule has never been reported to date. Studies are underway to further understand the origin of this dramatic transformation.

Biosynthesis remains a rich source of inspiration for discovering new strategies and tactics in chemical synthesis[39,40]. In the biosynthesis of the 3,4-dioxygenated 5-hydroxy-4-arylquinolin-2(1*H*)-one alkaloids, investigation of the aspoquinolones A and B[10], penigequinolones A and B and yaequinolone C[17,18], yaequinolones J1 and J2[26], biosynthetic pathways have revealed a number of unique mechanisms involved in the formation of this family of natural products[41–43]. An unprecedented mechanism of

iterative prenylation for installing the 10-carbon unit using two aromatic prenyltransferases (PenI and PenG) in the biosynthesis of this family of natural products has been identified[41]. Especially, Tang's group[42] discovered two Brønsted acid enzymes (PenF and AsqO) that can catalyze two unprecedented epoxide transformations that takes place under strong acid. Gratefully, several other members of this family natural products were synthesized by treating aniduquinolone A (**1**) with strong acid. The successful unprecedented acid-catalyzed rearrangement paves the way for the understanding of the biosynthetic relationships between aniduquinolones and aflaquinolones. And those natural products whose biogenetic pathways are unknown or have not been investigated in detail, synthetic chemistry can provide inspiration on future new biosynthetic pathways investigations.

## Conclusion
In summary, we have accomplished the first scalable total synthesis of (+)-aniduquinolone A (**1**) starting from the commercially available 2-methyl-5-nitrophenol in 13 steps (the longest linear sequence) with an overall 3.9% yield. Two sets of interesting examples of diastereoisomers (compounds **1/26**, **25/27**) with substantially overlapping NMR data were synthesized

**Fig. 6 Proposed mechanism.** The acid-catalyzed rearrangement from aniduquinolone A to 19-*epi*-aniduquinolone A and aflaquinolones.

and characterized unambiguously. Particularly, aflaquinolones A, C, and D (**4**–**6**) were obtained by treating (+)-aniduquinolone A (**1**) with TFA, and this dramatic transformation offered unprecedented insight into the biosynthetic pathways of aflaquinolones. Furthermore, the present synthetic strategy could inspire further advances in the synthesis of this family of natural products and undoubtedly lay a solid foundation to create analogue libraries. Biological studies and medicinal chemistry research of **1**, as well as its analogues are currently underway in our laboratory, which will be reported in due course.

## Methods
**General information**. For more details, see Supplementary Methods.

**Synthesis and characterization**. See Supplementary Note 1 and Supplementary Figs. 10–85 for NMR spectra.

**Comparison of NMR spectra of the natural products and synthetic products**. see Supplementary Tables 1–4.

## Data availability

The X-ray crystallographic coordinates for the structure of **1**, **14**, and **29** have been deposited at the Cambridge Crystallographic Data Centre (CCDC), under deposition number 2122872, 2122875, and 2122876. These data can be obtained free of charge from The Cambridge Crystallographic Data Centre via www.ccdc.cam.ac.uk/data_request/cif. Other data are available from the authors upon reasonable request. The CIF files of CCDC 2122872, CCDC 2122875, and CCDC 2122876 are also included as Supplementary Data 1–3.

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

## Acknowledgements

We thank Syngenta for the fellowship to Feng-Wei Guo. This work was supported by the Programme of National Natural Science Foundation of China (Nos. U1706210, 41776141, and 41322037), the Fundamental Research Funds for the Central Universities (No. 201841004), AoShan Talents Programme Supported by Pilot National Laboratory for Marine Science and Technology (Qingdao) (No. 2015ASTP-ES11), the Programme of Natural Science Foundation of Shandong Province of China (No. JQ201510), and the Taishan Scholars Programme, China (No. tsqn20161010).

## Author contributions

F.W.G. and X.F.M. contributed equally to this work; C.L.S. directed the project; C.L.S., F.W.G., and X.F.M. conceived the synthetic route; F.W.G., X.F.M., and Y.Q. conducted the synthetic work; F.W.G., X.F.M., M.Y.W., G.Y.C., C.Y.W., Y.C.G., and C.L.S. analyzed the results; C.L.S., Y.C.G., F.W.G., and X.F.M. wrote the manuscript.

## Competing interests

The authors declare no competing interests.
