## [Peer Review File · Communications Chemistry]

REVIEWERS' COMMENTS:

Reviewer #1 (Remarks to the Author):

This manuscript details a competently executed total synthesis of the title 3,4-dioxygenated 5-hydroxy-4-aryl-quinolin-2(1H)-one alkaloid as well as its acid-catalyzed rearrangement to three others. The work is logically presented although I feel that the authors would benefit significantly from the assistance of a native English speaker in terms of “polishing” the text.

A key step used in the synthesis of the 4-aryl-quinolin-2(1H)-one core of the title natural product (compound 1 of the manuscript) is an intramolecular and diastereoselective aldol reaction of a type used previously by, for example, Christmann et al (Org. Lett. 2018, 20, 7661 and 2020, 20, 675 – refs. 23 and 21, respectively, of the current manuscript). The terpenoid residue of the same target has been prepared using methodology reported by Marshall some time ago (Org. Lett. 2003, 5, 1931) but not referenced by the authors. A HWE reaction is used to connect the terpenoid-derived fragment to a precursor to the developing 4-aryl-quinolin-2(1H)-one ring system of target 1 with the aldol chemistry being “brought into play” shortly thereafter and with the terpenoid fragment serving as something like a chiral auxiliary and thus resulting in the formation the two diastereoisomeric aldol products 23 and 23a. It is not clear, however, precisely how these were separated from one another (in the Exptl section they are both assigned the same R_f values). All of the steps involved in the synthesis have been conducted on at least a gram scale while single-crystal X-ray analyses have been conducted on various key compounds and so lending some confidence to the stereochemical assignments made throughout the paper.

The other important aspect of this paper derives from the observations that the tetrahydrofuran-containing residue within compound 1 undergoes acid-catalysed rearrangement reactions leading, as shown in Figure 6 and depending upon the specific conditions employed, to either its epimer 25 or the cyclohexanone-containing and naturally-occurring isomers 4, 5 and 6.

Overall, this is a nice body of work but I think that it is better suited for publication as a full paper given the related studies already published on the synthesis of 3,4-dioxygenated 5-hydroxy-4-aryl-quinolin-2(1H)-ones.

Regardless of the ultimate destination of the paper, there are quite a number of typographical and other errors in the paper that will need addressing. Indicative ones are:

Figure 2: the term “intramolecular aldol cyclization” is tautological - all cyclizations are intramolecular processes;

Figure 3: First Step of Scheme – HMTA not HMPA;

Figure 6 Caption: What is an “acid-inspired intramolecular transformation”?? Surely “acid-catalyzed rearrangement” is the term that applies here!!

Figure 6: Strictly speaking the conversion of E into compounds 4, 5 and 6 involves enol- keto tautomerism (and not keto-enol tautomerism);

Line 208: The sentence “To our surprise, when (+)-aniduquinolone A(1) was treated with TFA in CH₂Cl₂ at room temperature, aflaquinolone A(4), aflaquinolone C (5) and aflaquinolone D (6) together with another diastereoisomer were obtained in 70% combined yield (Fig. 6).” implies four products were obtained from this reaction. What was the nature of the fourth compound?

Discussion: This should be re-labelled as a conclusion or summary;

Supplementary Table 3, Caption: Aflaquinolone A (not the plural) – the same issue applies elsewhere.

Reviewer #2 (Remarks to the Author):

The manuscript reports a total synthesis of (+)-aniduquinolone A which is scalable. It also discloses an acid-promoted transformation to aflaquinolones.

The manuscript can be considered as suitable for publication, provided the authors are willing to pay due attention to the following minor details:

- 1- Please correct typographic errors, such as Fernandez-Ibañez in Fig. 2a, as well as extra and missing spaces, etc.
- 2- Please ensure the spelling of Fernández-Ibañez is consistent (with n or ñ).
- 3- Please thoroughly revise the text; there are several errors related to the use of the English language (compound 27 was found to be differ from 25 that produced... And only a few edificatory examples of diastereoisomers???... which indicated the action mode of acid-inspired intramolecular transformation???)
- 4- Authors include a short account related to the evolution of the synthesis of these compounds. In their account they mention that "Christmann and co-workers found out that the 6-propenyl side chain could be assembled by Claisen rearrangement of 5-O-allyl in the heterocycle". They should note it was actually the work of Simonetti et al. (ref. 25) the one that suggested using a Claisen rearrangement to "assemble the 6-propenyl side chain". In order to make a true story, this detail and the corresponding reference (ref. 25) must be added to the text.
- 5- The authors used an acetate group to protect a phenol; it is known that this group is unable to withstand HWE conditions, where strong bases are used. Authors should discuss the need to protect 14 as an acetate instead of as a MOM ether derivative.
- 6- In the phrase "Especially, they discovered two Brønsted acid enzymes..." it is not clear who are they.
- 7- Author contributions: Use authors as a plural of author.

Reviewer #3 (Remarks to the Author):

This work reported the first scalable total synthesis of (+)-aniduquinolone A starting from the commercially available 2-methyl-5-nitrophenol in 13 LLS steps and 3.9% overall yield. The reported synthesis is well planned by a convergent approach through the simple but crucial stereoselective Horner-Wadsworth-Emmons (HWE) reaction, leading to a gram-scale crop finally. In addition, the unexpected chemistry of acid-promoted cationic transformations upon the THF-intermediates 23/23a and 24/24a allowed quick achievement of several natural aflaquinolones bearing a variable cyclohexanone subunit in a biomimetic fashion. In overall, this manuscript is recommended to publish on Commination Chemistry after proper minor revisions (see below).

MS page 6, Fig 3. Reaction conditions a), the reagent is not HMPA! It is hexamethylenetetramine (HMTA).

MS page 7, Table 1: E/Z ratio of 11 (even up to 100% selectivity) to be noted in the manuscript and table note.

MS page 8, line 126: as mentioned, the IM aldol provides two (23 and 23a, both syn-) of the four possible diastereomers. An explanation should be given on such a stereoselectivity.

SI, page 6: characterizations of SI-6, the ¹HNMR, rotation [α]_D, and ee value should be determined and reported.

Thank you very much for your support and for the reviewers' comments concerning our manuscript entitled "Scalable total synthesis of (+)-aniduquinolone A and its unexpected transformation to aflaquinolones with acid catalysis" (manuscript ID COMMSCHEM-21-0406-T). The suggestions and comments are all valuable and very helpful for revising and improving our manuscript. We have studied these suggestions and comments carefully and made efforts for the corrections. Revised portions are marked in red color in the "Revised Manuscript" file. The main corrections in the manuscript and reviewers' comments are as follows:

Reviewer #1:

This manuscript details a competently executed total synthesis of the title 3,4-dioxygenated 5-hydroxy-4-aryl-quinolin-2(1H)-one alkaloid as well as its acid-catalyzed rearrangement to three others. The work is logically presented although I feel that the authors would benefit significantly from the assistance of a native English speaker in terms of "polishing" the text.

A key step used in the synthesis of the 4-aryl-quinolin-2(1H)-one core of the title natural product (compound 1 of the manuscript) is an intramolecular and diastereoselective aldol reaction of a type used previously by, for example, Christmann et al (Org. Lett. 2018, 20, 7661 and 2020, 20, 675 – refs. 23 and 21, respectively, of the current manuscript). The terpenoid residue of the same target has been prepared using methodology reported by Marshall some time ago (Org. Lett. 2003, 5, 1931) but not referenced by the authors. A HWE reaction is used to connect the terpenoid-derived fragment to a precursor to the developing 4-aryl-quinolin-2(1H)-one ring system of target 1 with the aldol chemistry being "brought into play" shortly thereafter and with the terpenoid fragment serving as something like a chiral auxiliary and thus resulting in the formation the two diastereoisomeric aldol products 23 and 23a. It is not clear, however, precisely how these were separated from one another (in the Exptl section they are both assigned the same R_f values). All of the steps involved in the synthesis

have been conducted on at least a gram scale while single-crystal X-ray analyses have been conducted on various key compounds and so lending some confidence to the stereochemical assignments made throughout the paper.

The other important aspect of this paper derives from the observations that the tetrahydrofuran-containing residue within compound 1 undergoes acid-catalysed rearrangement reactions leading, as shown in Figure 6 and depending upon the specific conditions employed, to either its epimer 25 or the cyclohexanone-containing and naturally-occurring isomers 4, 5 and 6.

Overall, this is a nice body of work but I think that it is better suited for publication as a full paper given the related studies already published on the synthesis of 3,4-dioxygenated 5-hydroxy-4-aryl-quinolin-2(1H)-ones.

Response: Thank you very much for your important comments and nice suggestions! They are very valuable for improving our manuscript. We have studied these comments carefully and we are trying our best to solve these corrections which we hope meet with approval. All these corrections are as follows:

- (1) Professor Yu-Cheng Gu (Syngenta Jealott's Hill International Research Centre) has polished and proofread our manuscript.
- (2) We cited this report (Org. Lett. 2003, 5, 1931) in Retrosynthetic analysis part – refs 28) in the revised manuscript.
- (3) As you know, the polarities of the two products **23** and **23a** are very close. It is very challenging to separate and purify these two compounds by silica gel column chromatography, HPLC, or chiral HPLC. Fortunately, recrystallization presents an indispensable step in downstream separation and purification processing. Through the attempt of various solvents, the two diastereoisomeric aldol products **23** and **23a** can be separated from one another by recrystallization in MeOH.

Chiral HPLC analysis: Chiralpak IC column (5 μ m, 4.6 mm \times 250 mm); hexane/*i*-propanol = 85:15, 0.5 mL/min, λ = 254 nm; t_R [**23**] = 14.0 min, t_R [**23a**] = 26.0 min.

a: crystal, **23** (1st recrystallization). b: solution (1st recrystallization). c: crystal, **23** (2nd recrystallization).

1. Figure 2: the term “intramolecular aldol cyclization” is tautological - all cyclizations are intramolecular processes;

Response: Thank you so much for your reminding. The term “intramolecular aldol cyclization” has been revised to “intramolecular aldol reaction”.

2. Figure 3: First Step of Scheme – HMTA not HMPA;

Response: Thank you so much for your reminding. We have revised this mistake. “HMPA” was corrected to “HMTA”.

3. Figure 6 Caption: What is an “acid-inspired intramolecular transformation”?? Surely “acid-catalyzed rearrangement” is the term that applies here!!

Response: Thank you very much for your suggestions. The term “acid-inspired intramolecular transformation” has been revised to “acid-catalyzed rearrangement”.

4. Figure 6: Strictly speaking the conversion of E into compounds 4, 5 and 6 involves enol- keto tautomerism (and not keto-enol tautomerism);

Response: Thank you for your suggestions. The term “keto-enol tautomerism” has been revised to “enol- keto tautomerism”.

5. Line 208: The sentence “To our surprise, when (+)-aniduquinolone A (1) was treated with TFA in CH₂Cl₂ at room temperature, aflaquinolone A (4), aflaquinolone C (5) and aflaquinolone D (6) together with another diastereoisomer were obtained in 70% combined yield (Fig. 6).” implies four products were obtained from this reaction. What was the nature of the fourth compound?

Response: Thank you so much for the reviewer about the question of the fourth compound. There were four products (aflaquinolone A, aflaquinolone C, aflaquinolone D and 19, 21-*epi*-aflaquinolone D) obtained from acid-catalyzed rearrangement. The ¹H and ¹³C NMR data of the fourth compound (19, 21-*epi*-aflaquinolone D) were the same as those of aflaquinolone D (6). Therefore, 19, 21-*epi*-aflaquinolone D was identified as a diastereomer of aflaquinolone D (6). And it is possible that 19, 21-*epi*-aflaquinolone D is natural product which have not yet been discovered in nature. We have re-written the sentence as follows.

To our surprise, when (+)-aniduquinolone A (1) was treated with TFA in CH₂Cl₂ at room temperature, aflaquinolone A (4), aflaquinolone C (5), aflaquinolone D (6) and 19, 21-*epi*-aflaquinolone D (32) were obtained in 70% combined yield.

6. Discussion: This should be re-labelled as a conclusion or summary;

Response: Thank you for your reminding. We have re-labelled this title as a conclusion.

7. Supplementary Table 3, Caption: Aflaquinolone A (not the plural) – the same issue applies elsewhere.

Response: Thank you so much for your reminding. According to your advice, this manuscript and Supporting Information were checked again by the authors to make sure there were no language issues. These errors have been modified and marked according to your suggestions. Other changes have been highlighted in red in the revised manuscript.

Reviewer #2:

The manuscript reports a total synthesis of (+)-aniduquinolone A which is scalable. It also discloses an acid-promoted transformation to aflaquinolones.

The manuscript can be considered as suitable for publication, provided the authors are willing to pay due attention to the following minor details:

1. Please correct typographic errors, such as Fernandez-Ibañez in Fig. 2a, as well as extra and missing spaces, etc.

Response: Thanks for your reminding and suggestions. We have carefully revised numerous typographic errors in the manuscript.

2. Please ensure the spelling of Fernández-Ibañez is consistent (with n or ñ).

Response: Thank you so much for your reminding. We have carefully checked the spelling of Fernández-Ibañez in the manuscript.

3. Please thoroughly revise the text; there are several errors related to the use of the English language (compound 27 was found to be differ from 25 that produced... And only a few edificatory examples of diastereoisomers???... which indicated the action mode of acid-inspired intramolecular transformation???)

Response: Thank you so much for your suggestions. According to your advice, this manuscript was checked again by the authors to make sure there were no language issues. We have re-written the sentence as follows.

To our surprise, the two compounds **25** and **27** have different retention times and almost identical NMR data (Fig. 5). As far as we know, only a few edificatory examples of diastereoisomers with substantially identical ^1H and ^{13}C NMR data were reported³⁴⁻³⁸.

4. Authors include a short account related to the evolution of the synthesis of these compounds. In their account they mention that “Christmann and co-workers found out that the 6-propenyl side chain could be assembled by Claisen rearrangement of 5-O-allyl in the heterocycle”. They should note it was actually the work of Simonetti et al. (ref. 25) the one that suggested using a Claisen rearrangement to “assemble the 6-propenyl side chain”. In order to make a true story, this detail and

the corresponding reference (ref. 25) must be added to the text.

Response: Thank you so much for your reminding and suggestions. We have re-summarized the literature on the evolution of the synthesis of these compounds. We have re-written the sentences as follows and adjusted the order of references.

Simonetti and co-workers developed a convenient way to install 6-propenyl side chain by Claisen rearrangement of 5-*O*-allyl in the heterocycle²¹. Similar reaction was also used in the construction of unsaturated pyran fragments in yaequinolones J1 and J2 by Vece and co-workers²², as well as the synthesis of aniduquinolone C and peniprequinolone by Christmann's group²³. Recently, Christmann's group²⁴ developed the tandem Knoevenagel electrocyclization to further optimize the assembly of pyran fragments in yaequinolones J1 and J2. In 2021, the group of Fernández-Ibáñez²⁵ successfully synthesized yaequinolone-related natural products by late-stage C–H olefination to introduce C-6 side chains.

21. Simonetti, S. O., Larghi, E. L. & Kaufman, T. S. A convenient approach to an advanced intermediate toward the naturally occurring, bioactive 6-substituted 5-hydroxy-4-aryl-1*H*-quinolin-2-ones. *Org. Biomol. Chem.* **14**, 2625–2636 (2016).
22. Vece, V., Jakkepally, S. & Hanessian, S. Total synthesis and absolute stereochemical assignment of the insecticidal metabolites yaequinolones J1 and J2. *Org. Lett.* **20**, 4277–4280 (2018).
23. Schwan, J., Kleoff, M., Hartmayer, B., Heretsch, P. & Christmann, M. Synthesis of quinolinone alkaloids via aryne insertions into unsymmetric imides in flow. *Org. Lett.* **20**, 7661–7664 (2018).
24. Schwan, J., Kleoff, M., Heretsch, P. & Christmann, M. Five-step synthesis of yaequinolones J1 and J2. *Org. Lett.* **22**, 675–678 (2020).
25. Jia, W. L., Ces, S. V. & Fernández-Ibáñez, M. Á. Divergent total syntheses of yaequinolone-related natural products by late-stage C–H olefination. *J. Org. Chem.* **86**, 6259–6277 (2021).

5. The authors used an acetate group to protect a phenol; it is known that this group is unable to withstand HWE conditions, where strong bases are used. Authors should discuss the need to protect 14 as an acetate instead of as a MOM ether derivative.

Response: Thank you for your comments. It is really a good point. We tried many experiments with different protected groups. Before 14 was protected as an acetate,

the methoxymethyl (MOM)-protected **14** was used to undergo the radical-mediated bromination. To our disappointment, the methoxymethyl (MOM)-protected **14** gave a complex mixture under basic condition. This key information has been integrated into the article.

We have added the sentence as follows.

Pyridinium chlorochromate (PCC) oxidation²⁹ of secondary alcohol **19** afforded the corresponding ketone **14**. The structure of **14** was further confirmed by single-crystal X-ray analysis. The unprotected hydroxyl group in **14** was converted into a MOM ether derivative. The radical-mediated bromination of methoxymethyl (MOM)-protected **14** in the presence of NBS with a catalytic amount of AIBN failed to produce any desired product. Gratifyingly, the radical-mediated bromination of **20** under basic condition furnished bromoester that was reacted directly with an excess of triethyl phosphite and delivered the desired phosphonate **21** in 72% yield (15.0 g scale)³⁰.

6. In the phrase “Especially, they discovered two Brønsted acid enzymes...” it is not clear who are they.

Response: Thank you so much for your reminding. We have re-written the sentence as follows.

Especially, Tang’s group⁴² discovered two Brønsted acid enzymes (PenF and AsqO) that can catalyze two unprecedented epoxide transformations that takes place under strong acid.

7. Author contributions: Use authors as a plural of author.

Response: Thank you so much for your reminding. We have re-written the sentence as follows.

F. W. G. and X. F. M. contributed equally to this work; C. L. S. directed the project; C. L. S., F. W. G. and X. F. M. conceived the synthetic route; F. W. G., X. F. M. and Y. Q. conducted the synthetic work; F. W. G., X. F. M., M. Y. W., G. Y. C., C. Y. W., Y. C.

G. and C. L. S. analysed the results; C. L. S., Y. C. G., F. W. G. and X. F. M. wrote the manuscript.

Reviewer #3:

This work reported the first scalable total synthesis of (+)-aniduquinolone **A** starting from the commercially available 2-methyl-5-nitrophenol in 13 LLS steps and 3.9% overall yield. The reported synthesis is well planned by a convergent approach through the simple but crucial stereoselective Horner-Wadsworth-Emmons (HWE) reaction, leading to a gram-scale crop finally. In addition, the unexpected chemistry of acid-promoted cationic transformations upon the THF-intermediates **23/23a** and **24/24a** allowed quick achievement of several natural aflaquinolones bearing a variable cyclohexanone subunit in a biomimetic fashion. In overall, this manuscript is recommended to publish on *Commination Chemistry* after proper minor revisions (see below).

1. MS page 6, Fig 3. Reaction conditions a), the reagent is not HMPA! It is hexamethylenetetramine (HMTA).

Response: Thank you so much for your reminding. We have revised this mistake. “HMPA” was corrected to “HMTA” (page 6, Fig 3. Reaction conditions a).

2. MS page 7, Table 1: E/Z ratio of **11** (even up to 100% selectivity) to be noted in the manuscript and table note.

Response: Thank you for your valuable suggestions, they are very important for improving our researches and manuscript. In order to clarify this part, the *E/Z* ratio of **11** has been added in the manuscript and table note. We have re-written the sentence as follows.

Fortunately, we discovered eventually that MOM protected phosphonate **12** underwent olefination smoothly to obtain the desired product **11** in 78% yield with no detection of the corresponding *Z*-isomer (Table 1, entry 3, 3.9 g scale, *E/Z* = 100:0).

3. MS page 8, line 126: as mentioned, the IM aldol provides two (**23** and **23a**, both syn-) of the four possible diastereomers. An explanation should be given on such a stereoselectivity.

Response: Thank you so much for your suggestions. The corresponding paragraph and reference have been added to explain the highly diastereoselective intramolecular aldol reaction in the manuscript.

The more stable *Z*-enolate is presumably responsible for the observed diastereoselectivity³².

32. Ueki, H., Ellis, T. K., Khan, M. A. & Soloshonok, V. A. Highly diastereoselective synthesis of new, carbostyryl-based type of conformationally-constrained beta-phenylserines. *Tetrahedron*. **59**, 7301–7306 (2003).

4. SI, page 6: characterizations of SI-6, the ¹H NMR, rotation [α]_D, and ee value should be determined and reported.

Response: Thank you so much for your reminding. Compound SI-6 is a known compound that has been reported by Marshall's group. The ¹H NMR and ¹³C NMR data for compound SI-6 has been added in the Supporting Information.

Special thanks for your valuable comments. We tried our best to improve our manuscript. All these corrections will not change the framework of this manuscript, and we list the corrections and marked in red color in the revised manuscript.

Best wishes!

Sincerely,

Chang-Lun Shao